# The Effectiveness and Tolerability of a Very Low-Volume Bowel Preparation for Colonoscopy Compared to Low and High-Volume Polyethylene Glycol-Solutions in the Real-Life Setting

**DOI:** 10.3390/diagnostics12051155

**Published:** 2022-05-06

**Authors:** Olga Bednarska, Nils Nyhlin, Peter Thelin Schmidt, Gabriele Wurm Johansson, Ervin Toth, Perjohan Lindfors

**Affiliations:** 1Department of Gastroenterology, Linköping University Hospital, S-581 85 Linköping, Sweden; 2Department of Gastroenterology, Faculty of Medicine and Health, Örebro University, S-701 85 Örebro, Sweden; 3Department of Medicine, Ersta Hospital, S-116 91 Stockholm, Sweden; peter.thelin.schmidt@erstadiakoni.se or; 4Department of Medicine, Karolinska Institutet, S-171 77 Solna, Sweden; 5Department of Gastroenterology, Skåne University Hospital, Lund University, S-205 02 Malmö, Sweden; gabriele.wurmjohansson@skane.se (G.W.J.); ervin.toth@med.lu.se (E.T.); 6Department of Clinical Neuroscience, Division of Psychology, Karolinska Institutet, S-171 77 Solna, Sweden; perjohan.lindfors@ki.se or; 7Aleris Gastromottagningen City, S-111 37 Stockholm, Sweden

**Keywords:** bowel preparation, colonoscopy, polyethylene glycol, polyethylene glycol plus ascorbate, effectiveness, tolerability

## Abstract

Adequate bowel cleansing is essential for high-quality colonoscopy. Recently, a new very low-volume 1 litre (1L) polyethylene glycol (PEG) plus ascorbate solution (ASC) has been introduced. Our aims were to assess the effectiveness and tolerability of this product compared to low-volume 2L PEG-ASC and high-volume 4L PEG solutions, in a real-life setting. In six endoscopy units in Sweden, outpatients undergoing colonoscopy were either prescribed solutions according to local routines, or the very low-volume solution in split dose regimen. Bowel cleansing effectiveness and patient experience was assessed using the Boston Bowel preparation scale (BBPS) and a patient questionnaire. A total of 1098 patients (mean age 58 years, 52% women) were included. All subsegment and the total BBPS scores were significantly greater for 1L PEG-ASC in comparison to other solutions (*p* < 0.05 for 1L PEG-ASC and 4L PEG for transverse and left colon, otherwise *p* < 0.001). Nausea was more frequent with 1L PEG-ASC compared to 2L PEG-ASC (*p* < 0.001) and vomiting were more often reported compared to both other solutions (*p* < 0.01 and *p* < 0.05 for 2L PEG-ASC and 4L PEG, respectively). Smell, taste, and total experience was better for 1L PEG-ASC compared to 4L PEG (*p* < 0.001), and similar compared to the 2L PEG-ASC. In conclusion, 1L PEG-ASC leads to better bowel cleansing compared to 2L PEG-ASC or 4L PEG products, with similar or greater patient satisfaction.

## 1. Introduction

High-quality colonoscopy has proved to reduce both the incidence and mortality of colorectal cancer when applied in the general population [1,2,3]. Adequate bowel preparation assures accuracy of the colonoscopy examination. Meta-analyses show significant association between inadequate bowel preparation and decreased adenoma detection rates (ADR) including advanced adenomas [4,5], as well as threefold greater miss rate for ≥5 mm sized adenomas [6]. In fact, for subtle, flat lesions such as sessile serrated polyps the detection rates are negatively impacted by even a small decrease in bowel preparation quality below high-quality [7].

Additionally, caecal intubation rate [8,9,10] and satisfactory patient experience [11] are shown to be significantly correlated to adequate bowel cleansing. Poor bowel preparation results in re-scheduling of the procedure, which increases healthcare costs [12,13]. Therefore, the limit of ≥90%, with a quality target of ≥95% of adequate bowel preparation assessed using validated scales has been one of the most important quality indicators according to the European Society of Gastrointestinal Endoscopy (ESGE) [14]. The recommended split-dose regimen for the routine bowel preparation, shown to improve the quality of bowel cleansing [15,16], divides the amount of the liquids to be consumed in two parts. Nevertheless, the high-volume 4 litre (L) polyethylene glycol (PEG) solution to be ingested as 2 L at each dose results in diminished patients’ compliance [17]. There are two different high-volume PEG solutions on the market in Sweden, one containing sulphate (Laxabon) and another sulphate free and with citrus taste (Vistaprep). Both preparations showed the same cleansing effectiveness, while the sulphate free version has been proven to be superior in smell, taste, and total experience of laxation. However, no significant difference in patients’ compliance was seen [18,19]. In order to reduce the volume of the PEG solution to be consumed, different adjuvants can be used, as for instance high doses of ascorbate added to 2 L PEG (2L PEG-ASC, Movprep). In some older studies, no differences in efficacy or patient compliance between 2L PEG-ASC and 4L PEG could be shown [20,21] despite the proven better palatability [20] of the 2L PEG-ASC. However, a recent metanalysis demonstrated greater acceptability, better compliance, and willingness to repeat the same regime of 2L PEG-ASC versus 4L PEG [22].

Recently, a very low-volume 1 litre PEG solution with adjuvant ascorbate (1L PEG-ASC, Plenvu, Norgine) has been introduced in Europe and has shown to be at least noninferior to both low-volume 2L PEG-ASC and high-volume 4L PEG solutions [23,24,25,26,27,28,29,30,31,32], with a favourable patient experience, tolerability, and high adherence [23,28,33].

The aim of our study is to validate the effectiveness and tolerability of very low-volume 1L PEG-ASC, newly established at the Swedish market, against the widely used high-volume and low-volume PEG solution in real-life clinical settings.

## 2. Materials and Methods

We designed a qualitative, prospective, multicentre, comparative, single-blinded, observational study that was performed at six Swedish Gastroenterology and Endoscopy units in out-patients prepared with high-volume or low-volume versus very low-volume PEG solution as bowel preparation before colonoscopy. The study was approved by the Swedish Ethical Review Authority (2020-05983) and registered in an international clinical trials registry (ClinicalTrials.gov ID NCT05192551). All collected patient data during the study were unidentified.

Adult out-patients undergoing screening, surveillance, or diagnostic colonoscopy, with intention to conduct a complete colonoscopy, were consecutively enrolled from both the external (referred from another health centre) and internal (referred from the outpatient clinic at the same centre) referral lists at the study centres. The exclusion criteria were age below 18 years and illegibility. The patients were consecutively assigned to one of the two study groups: the standard group (bowel preparation with 2L PEG-ASC or 4L PEG solutions according to local routines) and the very low-volume group (bowel preparation with 1L PEG-ASC), aiming at approximately 100 patients in each group. The colonoscopists were blinded to the study product used. The overnight split-dosing bowel cleansing regimen was applied throughout the study. For both the 2L PEG-ASC and the very low-volume PEG solution preparation included 500 mL additional clear fluids after each dose, apart from additional clear fluids “ad libitum” that were recommended up to two hours before the procedure in both study arms. A low-fibre diet was recommended one week before the procedure, while on the day before the colonoscopy light breakfast and lunch were permitted.

The study was conducted according to standard procedures with no additional intervention apart from the pre-procedural patient questionnaire including information on age, gender, and patient’s personal experience with the study preparation (smell, taste, total volume of laxative ingested, total fluid ingested, total experience) using a 5-point Likert scale as well as tolerability regarding nausea and vomiting. Patients were asked to attend the study at the time of arrival to the endoscopy unit. If they accepted, an informed consent was signed, and the patient answered the pre-procedural questionnaire.

Experienced colonoscopists, blinded to treatment allocation, performed all colonoscopies and allocated initial segmental cleansing scores using the Boston Bowel Preparation Score [34]. Adequate bowel cleansing was defined as a total BBPS ≥ 6 with a partial BBPS ≥ 2 in each colon segment. The right-sided and total high-quality bowel cleansing were defined as BBPS Right colon = 3 and total BBPS = 9, respectively. The information about BBPS score, caecal intubation rate, the need and reason for re-scheduling of the colonoscopy assessed by the colonoscopist, as well as the laxative used, were documented at the patient’s questionnaire by the endoscopy staff and collected without notifying the colonoscopist.

Results are presented as mean values with standard deviations, except for the Likert-scale results, where medians and percentiles are reported. One-way ANOVA and Tukey post-hoc analyses were used for comparisons with continuous variables, Kruskal–Wallis one-way ANOVA with Bonferroni corrections were used for ordinal variables and Pearson Chi-Square test was used for categorical variables. Multivariate and univariate Logistic regression models were constructed to assess predictors for the outcomes: (a) adequate bowel cleansing (BBPS total score ≥ 6), (b) high-quality bowel cleansing (BBPS total score = 9) and (c) high-quality cleansing of the right colon (BBPS Right colon = 3). Covariates were selected based on their potential biological association with this outcome. The following covariates were included in the regression models: age (>65 or <65); gender (male or female), and type of bowel preparation (1L PEG-ASC, 2L PEG-ASC, or 4L PEG). In addition, sensitivity analyses were performed with the following covariates: age categorized into quartiles, ingestion of the entire amount of laxative, drinking of 1 litre or more of additional fluids or report of vomiting. All tests were two-tailed, and *p*-values of <0.05 were considered statistically significant. Statistical analyses were performed using SPSS Statistics for Windows version 27.0 (IBM Corp., Armonk, NY, USA, 2020).

## 3. Results

### 3.1. Study Population and Characteristics

A total of 1121 patients were included in the study. However, for 23 patients, information on the bowel preparation agent was missing and these patients were therefore excluded from the analyses. For the remaining 1098, mean age was 58 years, 52% women and 48% men. Four patients did not report their age and 45 patients did not report their gender.

The endoscopy units participating in this study included three university hospitals (Linköping, Malmö, and Örebro), one local hospital (Karlskoga), and two large endoscopy units in Stockholm with mainly outpatients referred from primary care (Stockholm Ersta and Stockholm GMC-Aleris Gastromottagning City).

No statistical differences in patients’ characteristics were seen between the different endoscopy units regarding gender, but age differed. Patients from Stockholm GMC were significantly younger compared to all other endoscopy centres apart from Linköping, and Linköping’s patients were younger than those from Karlskoga and Malmö (Table A1, Appendix A).

No statistical difference in bowel cleansing efficacy was observed between the two different high-volume PEG-solutions (Vistaprep and Laxabon), nor were there any significant differences in gender, caecal intubation rates, nausea rates, or vomiting rates. Smell was graded better for Vistaprep compared to Laxabon (*p* = 0.011), but not taste or total experience. Moreover, Vistaprep patients were significantly younger than Laxabon patients: mean 56 years compared to 63 (*p* < 0.001).

### 3.2. Bowel Cleansing Efficacy

BBPS total score was calculated in 1059 of the cases, i.e., if all three subsegments were reported. All BBPS subsegment scores and total score were significantly better for the 1L PEG-ASC group compared to both 2L PEG-ASC and 4L PEG groups (Table 1 and Figure 1).

Adequate cleansing, as defined by a BBPS score of ≥6 with no segment less than 2, was overall very good and was achieved in 97% of cases in the 1L PEG-ASC and 4L PEG group and 95% of cases in the 2L PEG-ASC group. There were no statistical differences between the groups. When a total score of 9 was used to define a high-quality cleansing of the colon, there was a significantly better result in the 1L PEG-ASC compared to both 2L PEG-ASC and 4L PEG groups. High-quality cleansing of the right colon, defined as a BBPS subscore of 3, was also seen significantly more often in the 1L PEG-ASC compared to both 2L PEG-ASC and 4L PEG groups.

Caecal intubation rate was 95% for the 4L PEG group and the 1L PEG-ASC group, whereas for the 2L PEG-ASC group it was 90%.

A total of 61 colonoscopies were reported to be incomplete. The reason was inadequate cleansing in 21 cases, technical difficulties in 21 cases, and for the remaining 19, “other reason” was reported as the cause, including patients with previous right sided hemicolectomy, finding of a tumour or inflammation. There were no statistically significant differences between the different treatment groups for reasons of incompletion.

### 3.3. Predictors of Overall Cleansing Success and High-Quality Cleansing of the Right Colon

Univariate and multivariate logistic multiple regression models for overall adequate cleansing success and high-quality cleansing of total colon and the right colon were created with predictors set to bowel preparation, gender, and age categorized into two cohorts: younger than 65 or 65 and older. Including additional predictors such as age categorized into quartiles, ingestion of the entire amount of laxative, drinking of 1 L or more of additional fluids, or vomiting did not change the multivariate results.

As shown in Table 2A, the odds ratios for adequate bowel cleansing were lower, both in the univariate and multivariate analyses if the patient was over 65 years of age. Apart from that, no other significant predictor for adequate bowel cleansing was identified. For high-quality bowel cleansing, there were lower odds ratios both for the 2L PEG-ASC and 4L PEG groups compared to the 1L PEG-ASC group (Table 2B). Similarly, the odds ratios for high-quality cleansing of the right colon were lower for both the 2L PEG-ASC and 4L PEG groups compared to the 1L PEG-ASC group (Table 2C). Age ≥ 65 was also a negative predictive factor for high-quality colon cleansing, for both total score and in the right colon.

### 3.4. Adherence and Tolerability

Data on adherence and tolerability are presented in Table 1 and Figure 2. There were significant differences between the treatment groups regarding adherence (ingestion of all the bowel preparation). Complete intake of the solution was most often reported in the 2L PEG-ASC group, and the smallest proportion was ingested in the 4L PEG group. Similarly, ingestion of 1 litre or more of additional fluids was most often reported in the 2L PEG-ASC group and least often in the 4L PEG group.

Nausea was significantly more commonly reported in the 1L PEG-ASC group compared to the 2L PEG-ASC group but did not significantly differ from the 4L PEG group, whereas vomiting was more common in the 1 L PEG-ASC group compared to both 2L and 4L groups.

The patients graded smell, taste, and total experience of the cleansing process from 1 to 5 in the patient enquiry. There was no significant difference between the 1L PEG-ASC and 2L PEG-ASC groups regarding smell, taste, or total experience, whereas significantly lower scores were seen for these parameters in 4L PEG group compared to the other groups.

## 4. Discussion

The present study is the first head-to head post-marketing study in Scandinavia of very low-volume 1L PEG-ASC with other well-established low-volume 2L PEG-ASC and high-volume 4L PEG solutions. This prospective, multicentre audit illustrates several important aspects relevant to practicing colonoscopists. In our study, 1L PEG-ASC, 2L PEG-ASC, and 4L PEG groups achieved adequate bowel cleansing, exceeding the minimum standard of 90% set by ESGE and meeting the quality target limit of ≥95% [14]. Statistically greater total BBPS scores as well as subsegment scores were demonstrated for 1L PEG-ASC compared to other PEG products. Moreover, the odds ratios for total high-quality bowel cleansing (BBPS = 9) were greater for 1L PEG-ASC compared to both 2L PEG-ASC and 4L PEG products. The only identifiable risk factor for inadequate bowel cleansing, low total high-quality, and low right-sided high-quality bowel preparation, was age ≥65 years old. The adherence for ingesting the entire amount of laxative was lower in the 1L PEG-ASC compared to the 2L PEG-ASC group, but greater than the 4L PEG group. This could possibly be explained by the fact that nausea and vomiting were more frequently reported in the 1L PEG-ASC group. Nevertheless, patient satisfaction was as high or higher for 1L PEG-ASC compared to 2 L PEG-ASC and 4 L PEG preparations.

During colonoscopy, complete visualization of the entire mucosa is one of the most important end points of this procedure, and it is associated with lower incidence of colorectal cancer [35,36]. BBPS is a validated scale for bowel cleansing evaluation [34], measuring bowel cleanliness after completing all the cleansing manoeuvres, and therefore it is considered to express the real-world way colonoscopy is performed [37]. An adequate level of bowel cleanliness, that is sufficient for identification of polyps >5 mm, has been established as total BBPS ≥ 6 [6]; however, high-quality colon cleansing (total BBPS score 7–9 or segmental BBPS = 3) proved to increase adenoma detection and is considered to be a clinical priority in pre-colonoscopy bowel preparation [38]. Attaining adequate cleansing in the right colon is proven to be more difficult to achieve than in the distal colon [39]. Our results of 1L PEG-ASC cleansing effectiveness are in line with other studies, showing at least noninferiority of 1L PEG-ASC compared to both low-volume 2L PEG-ASC and high-volume 4L PEG solutions [23,24,25,26,27,28,29,30,31,32]. Moreover, the odds ratios for high-quality colon cleansing, including high-quality right-sided colon preparation, were greater in our study for 1L PEG-ASC compared to other PEG products, reproducing the results from other studies [23,28,32,40,41].

Bowel preparation is known to be one of the most worrisome factors for patients scheduled for colonoscopy [42,43]. Theoretically, diminishing the volume of the bowel preparation formulation should result in improved patient compliance and acceptability due to the lower volume of fluid to be consumed. In several studies, the 1L PEG-ASC demonstrated favourable patient experience, tolerability, and high adherence [23,25,26,28,33,44]. In our study, patient compliance for laxative ingesting was superior for 1L PEG-ASC versus high-volume PEG but inferior to 2L PEG-ASC. A meta-analysis by Maida et al., 2020 [45] showed a lower tolerability profile of 1L PEG-ASC in comparison to 2L PEG, mostly due to nausea and vomiting, although this was not reproduced in the following real-life study conducted by the same research group [23].

The lower volume of 1L PEG-ASC causes greater osmolality load, further increased by a ten-fold higher dose of ascorbate, compared to 2L PEG-ASC, applied in only 500 mL of fluid, which presumably contributed to the observed increased nausea and vomiting reported in our study [46]. However, hyperosmolality is a plausible cleanliness drive, effectuating high quality bowel preparation, particularly in the proximal colon [47]. Hence the factor probably causing poorer adherence is at the same time feasibly resulting in improved bowel cleanliness. Poor palatability is a major drawback of all bowel preparation solutions and a significant obstacle as far as the very low volume-product is concerned. It was our experience when conducting this study that the patients reported a significantly worse taste of the second dose of the very low volume-preparation, presumably due to the pungent artificial flavour added to minimize the chemical taste of ascorbate. Chilling the solution as well as ingestion of more free water may help to circumvent both poor palatability and nausea. Interestingly, in our study nausea was more frequently reported in the 1L PEG-ASC group than in the 2L PEG-ASC group, but did not reach statistical significance versus the 4L PEG preparate not containing ascorbate, which could suggest complexity of this subjective symptom. Despite more frequent side effects, which likely resulted in the observed lower adherence, patient satisfaction of 1L PEG-ASC was at least not inferior to other study products. As suggested previously [47], this spurious inconsistency suggests that patient tolerance might be considerably better for the very low volume-product than for the standard products, provided comparable side effects, as nausea and vomiting are a serious impediment to patients’ compliance. Moreover, patient assessment, based on evaluation of only one type of bowel cleansing product, was per se not objective. Oliviera et al. demonstrated that more than 75% of the patients in their study who took both preparations preferred 1L PEG-ASC to 2L PEG-ASC [48]. In addition, Maida at al. confirmed using multiple regression that 1L PEG-ASC is an independent predictor of patient tolerance over standard PEG solutions, both low and high volume [23], which further strengthens the hypothesis that lower volume of the bowel preparation may improve adherence.

Our study has several strengths. This is a prospective, real-life study conducted both in tertiary centre-settings and non-university centres, ensuring sufficient diversity of patients and resulting in the adequate generalizability of the study results to the outpatient settings. The colonoscopists were all well-trained and blinded for the bowel preparation, avoiding the obstacle of individual bias on subjectivity in evaluation of bowel preparation quality. The split dose regimen, applied in all study arms as recommended by the recent ESGE guidelines [49], has consistently been considered to improve the quality of bowel preparation [15,16], and therefore represents one of the benefits of the study.

An important limitation of the study was the lack of randomization as well as not considering other potential patient-related risk factors of poor bowel preparation, such as history of constipation, comorbid conditions, body mass index, and medications [50]. Nonetheless, the lack of randomization is secondary to the nature of the study, which was designed to be observational and thus clinically more relevant. Furthermore, one of the inclusion criteria for the study was legibility of the Swedish language, which may have resulted in certain selection bias. Although both the patients, the endoscopy staff, and the colonoscopists were instructed not to discuss the individual’s bowel preparation regimen, this might be a potential source of bias as the process was not possible to be fully controlled by the investigators. Moreover, the study design did not account for the comparable number of endoscopies conducted by each colonoscopist in each study arm, which could result in differences in BBPS scoring [37]. Finally, there were significant age differences between the study sites.

In summary, this multicentre study of more than one thousand outpatients proved superiority in bowel cleanliness of very low-volume 1L PEG-ASC compared to low-volume 2L PEG-ASC and high-volume 4L PEG products, as far as both total BBPS scores and most importantly the right-sided subsegment BBPS scores are concerned. The total patient satisfaction was as great as or greater with very low-volume versus other PEG formulations, despite more frequent nausea and vomiting caused by the very low-volume product. Future developments should focus on improving the palatability and reducing the nausea and vomiting side-effects of the bowel preparation formations.

## Figures and Tables

**Figure 1 diagnostics-12-01155-f001:**
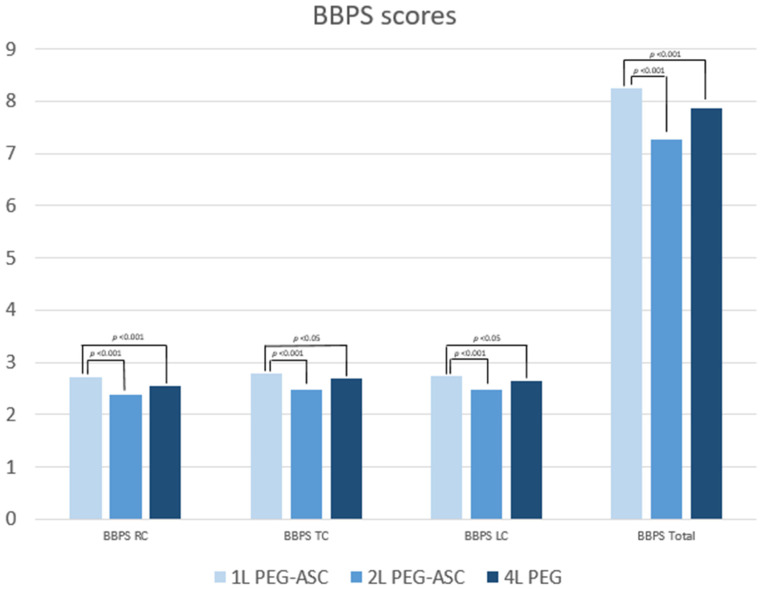
Boston Bowel Preparation Scores in the right colon (BBPS RC), the transverse colon (TC), and the left colon (LC), and BBPS total score (BBPS Total) for all study groups: 1L PEG-ASC (light blue), 2L PEG-ASC (blue), and 4L PEG (dark blue). PEG, polyethylene glycol; PEG-ASC, polyethylene glycol + Ascorbate.

**Figure 2 diagnostics-12-01155-f002:**
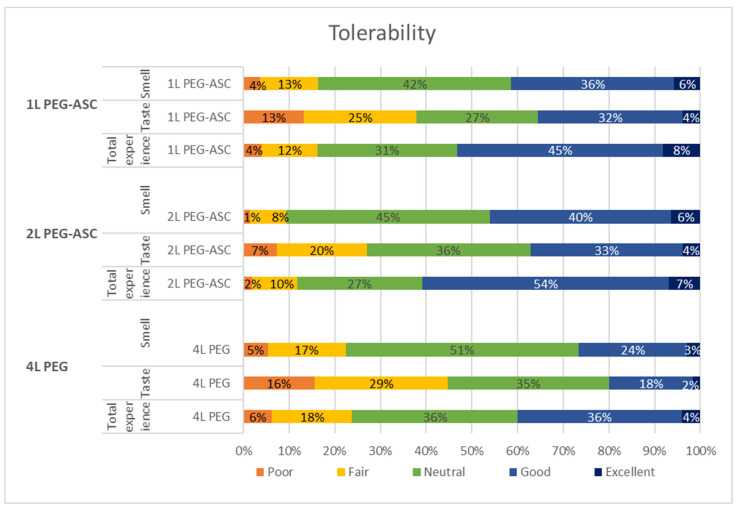
Stacked bar chart presenting proportions of answers for smell, taste, and total experience for the different laxatives studied, graded from poor to excellent. PEG, polyethylene glycol; PEG-ASC, polyethylene glycol + Ascorbate.

**Table 1 diagnostics-12-01155-t001:** Main results.

	Overall (n 1098)	1L PEG-ASC (n 523)	2L PEG-ASC (n 204)	4L PEG (n 371)	*p*-Value 1L vs. 2L PEG-ASC	*p*-Value 1L PEG-ASC vs. 4L PEG
Age (mean, min-max, SD)	58 (18–91, 16.9)	57 (18–91, 17.4)	60 (19–89, 16.4)	59 (19–90, 16.5)	**0.034**	0.241
Age < 65 year	612 (56%)	301 (58%)	105 (52%)	*206 (56%)*	0.212	0.381
Age ≥ 65 year	482 (44%)	219 (44%)	98 (48%)	165 (44%)	0.232	0.356
Male	503 (48%)	230 (46%)	103 (54%)	170 (47%)	0.051	0.618
Smell (median, pctl)	3.0 (3.0–4.0)	3.5 (3.0–4.0)	3.0 (3.0–4.0)	3.0 (3.0–4.0)	0.342	**<0.001**
Taste (median, pctl)	3.0 (2.0–4.0)	3.0 (2.0–4.0)	3.0 (2.0–4.0)	3.0 (2.0–3.0)	0.647	**<0.001**
Overall experience (median, pctl)	4.0 (3.0–4.0)	4.0 (3.0–4.0)	4.0 (3.0–4.0)	3.0 (3.0–4.0)	0.065	**<0.001**
Ingestion of all bowel preparation	978 (90%)	474 (91%)	195 (96%)	312 (85%)	**0.037**	**0.003**
≥1L additional fluids	691 (63%)	371 (71%)	164 (81%)	159 (43%)	**0.009**	**<0.001**
Nausea	404 (37%)	223 (43%)	45 (22%)	136 (37%)	**<0.001**	0.059
Vomiting	92 (8%)	60 (12%)	8 (4%)	11 (5%)	**0.002**	**0.011**
BBPS Right (mean, SD)	2.6 (0.6)	2.7 (0.5)	2.4 (0.7)	2.6 (0.6)	**<0.001**	**<0.001**
BBPS Transverse (mean, SD)	2.7 (0.5)	2.8 (0.5)	2.5 (0.7)	2.7 (0.5)	**<0.001**	**0.018**
BBPS Left (mean, SD)	2.7 (0.6)	2.8 (0.5)	2.5 (0.6)	2.7 (0.5)	**<0.001**	**0.015**
BBPS Total (mean, SD)	7.9 (1.6)	8.3 (1.5)	7.3 (2.0)	7.9 (1.4)	**<0.001**	**<0.001**
BBPS ≥ 6	1062 (97%)	508 (97%)	193 (95%)	361 (97%)	0.100	0.877
BBPS = 9	642 (61%)	370 (73%)	84 (45%)	188 (53%)	**<0.001**	**<0.001**
BBPS Right = 3	692 (66%)	391 (77%)	95 (51%)	206 (58%)	**<0.001**	**<0.001**
Cecal intubation rate	1020 (94%)	489 (95%)	177(90%)	354 (95%)	**0.024**	0.576
Incomplete due to inadequate laxation	21 (1.9)	11 (2.1%)	3 (1.5%)	7 (1.9%)	0.577	0.820
Incomplete due to technical reasons	21 (1.9)	8 (1.5%)	5 (2.5%)	8 (2.2%)	0.400	0.486
Incomplete due to other reasons	19 (1.7)	5 (1.0%)	12 (5.9%)	2 (0.5%)	**<0.001**	0.486

PEG, polyethylene glycol; PEG-ASC, polyethylene glycol + Ascorbate; BBPS, Boston Bowel Preparation Score; SD, standard deviation; pctl, 25–75th percentile.

**Table 2 diagnostics-12-01155-t002:** (**A**) Predictors for adequate bowel cleansing (BBPS total score ≥ 6). (**B**) Predictors for high-quality bowel cleansing (BBPS total score = 9. (**C**) Predictors for high-quality cleansing of the right colon (BBPS Right colon = 3).

(A)				
	Univariate		Multivariate	
	OR (95% CI)	*p*-Value	OR (95% CI)	*p*-Value
Age ≥ 65	0.34 (0.16–0.69)	**0.003**	0.36 (0.17–0.75)	**0.006**
Gender (male)	1.57 (0.78–3.15)	0.206	0.61 (0.30–1.23)	0.166
1L PEG-ASC	Ref		Ref	
2L PEG-ASC	0.52 (0.23–1.15)	0.105	0.51 (0.23–1.14)	0.099
4L PEG	1.07 (0.47–2.4)	0.877	1.20 (0.52–2.80)	0.667
**(B)**				
	**Univariate**		**Multivariate**	
	OR (95% CI)	*p*-Value	OR (95% CI)	OR (95% CI)
Age ≥ 65	0.73 (0.57–0.93)	**0.012**	0.74 (0.57–0.97)	**0.027**
Gender (male)	1.18 (0.92–1.53)	0.192	1.13 (0.87–1.47)	0.351
1L PEG-ASC	Ref		Ref	
2L PEG-ASC	0.30 (0.21–0.42)	**<0.001**	0.31 (0.22–0.44)	**<0.001**
4L PEG	0.41 (0.31–0.54)	**<0.001**	0.42 (0.31–0.56)	**<0.001**
**(C)**				
	**Univariate**		**Multivariate**	
	OR (95% CI)	*p*-Value	OR (95% CI)	*p*-Value
Age ≥ 65	0.74 (0.58–0.96)	**0.024**	0.77 (0.59–1.01)	0.057
Gender (male)	1.14 (0.88–1.48)	0.311	1.10 (0.83–1.42)	0.541
1L PEG-ASC	Ref		Ref	
2L PEG-ASC	0.30 (0.21–0.43)	**<0.001**	0.31 (0.22–0.45)	**<0.001**
4L PEG	0.41 (0.30–0.54)	**<0.001**	0.41 (0.30–0.55)	**<0.001**

BBPS, Boston Bowel Preparation Score; PEG, polyethylene glycol; PEG-ASC, polyethylene glycol + Ascorbate; OR, odds ratio; CI, confidence interval.

## Data Availability

The data presented in this study are available on request from the corresponding author (N.N.). The data are not publicly available due to privacy reasons.

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
