# Peer review of "The Effectiveness and Tolerability of a Very Low-Volume Bowel Preparation for Colonoscopy Compared to Low and High-Volume Polyethylene Glycol-Solutions in the Real-Life Setting"

_diagnostics, 2022, doi:10.3390/diagnostics12051155_

Round 1

Reviewer 1 Report

Summary: This is a well-designed study and appropriately written manuscript comparing the efficacy and tolerability of very low volume PEG-ASC solution to low volume and high volume PEG solutions. The authors found that very low volume solution was superior to the other PEG solutions based on BBPS scores (total and right-sided). Side effects including nausea and vomiting were more frequent in the very low volume solution.

My primary concern is the failure to adequately address the most obvious setback of the very low volume 1L PEG-ASC, which is its side effects profile. The solution was characterized by more frequent nausea and vomiting compared to the low volume and high volume PEG solutions. I feel that the explanation provided by the authors in lines 243-246 is somewhat limited and not reassuring. What are the authors thoughts on how the palatability can be improved?

The authors report that the adherence to the very low volume 1L PEG-ASC was lower compared to the 2L PEG-ASC solution, presumably due to the side effects of nausea and vomiting. How then do the authors reconcile the finding that patient satisfaction was as high or higher with very low-volume versus other PEG formulations? Adherence seems to be a more reliable marker of patient satisfaction.

Also if adherence to the very low volume solution was lower than the 2L PEG-ASC, how do the authors explain the better BBPS scores recorded by the very low volume solution compared to the 2L PEG-ASC?

Author Response

We thank the Reviewer for the constructive criticism and valuable comments which we have addressed to the best of our ability in the revised manuscript, and which are described point by point below. We believe that the revisions have considerably improved the manuscript.

Point-by-point responses to Reviewer 1 

My primary concern is the failure to adequately address the most obvious setback of the very low volume 1L PEG-ASC, which is its side effects profile. The solution was characterized by more frequent nausea and vomiting compared to the low volume and high volume PEG solutions. I feel that the explanation provided by the authors in lines 243-246 is somewhat limited and not reassuring. What are the authors thoughts on how the palatability can be improved?

Thank you for your important comment. We have adjusted the discussion in accordance with it on page 9, line 257-70.

The authors report that the adherence to the very low volume 1L PEG-ASC was lower compared to the 2L PEG-ASC solution, presumably due to the side effects of nausea and vomiting. How then do the authors reconcile the finding that patient satisfaction was as high or higher with very low-volume versus other PEG formulations? Adherence seems to be a more reliable marker of patient satisfaction.

We agree that this is a relevant consideration and we have added some lines regarding this spurious inconsistency in the Discussion on page 9, line 271-280.

Also if adherence to the very low volume solution was lower than the 2L PEG-ASC, how do the authors explain the better BBPS scores recorded by the very low volume solution compared to the 2L PEG-ASC?

This is an excellent point that we aimed to address in the Discussion on page 9, line 260-263.

Reviewer 2 Report

I uploaded a comment file. 

Author Response

We thank the Reviewer for the constructive criticism and valuable comments which we have addressed to the best of our ability in the revised manuscript, and which are described point by point below. We believe that the revisions have considerably improved the manuscript.

Point-by-point response:

BPPS, the typo of BBPS was repeatedly used throughout this manuscript including methods, discussion, Table 1, Table 2, and appendix Table. All BPPS need to be corrected.

Thank you for notifying us on this lapsus calami. All BPPS are corrected.

Abstract - line 32

Results of the abstract (line 32) and results of the main text (line 191) do not match.

Nausea in the 1L PEG-ASC group did not differ from the 4L PEG group. Thus, abstract needs to be corrected as follows.

Vomiting was more frequent with 1L PEG-ASC compared to other products. (line 32)

Thank you for recognizing this. The abstract is corrected.

Methods – line 94

Did 2L PEG-ASC regimen also include additional clear fluids after each dose? If so, it needs to be mentioned in the Methods section how much additional clear fluids were included in the 2L PEG-ASC regimen.

Yes, additional clear fluids was recommended for the 2L PEG-ASC regimen. We have added that information in the manuscript to clarify this.

Methods – statistics

The description of the logistic regression models is too simple. You need to be more specific.

In the multivariate analyses, which variables were included or adjusted for? Details need to be described in the Methods and Results section. It would be better to consult a biomedical statistician to describe statistical methods.

The description of the logistic regression models is now described in detail, after consulting a statistician.

Results

If you have data on serum electrolytes, creatinine results, and patients who complained of dizziness on the day of colonoscopy, and blood pressure, it would be better to show that data.

We did not include laboratory tests or blood pressure in the study design, so we do not have these data. However, previously published paper from Radaelli et al on safety data from more than 1000 patients receiving plenvu showed no major safety issues. Median changes in electrolytes and renal function from baseline were mild, transient and without any clinical relevance. However, serum sodium levels higher than normal were seen in a substantial number of patients and this might suggest that patients need to be instructed to take in more free water in order to prevent hypernatremia.

Results – line 134

centra à centre

Corrected.

Results – line 135

I think this is not Table 2, but Appendix Table.

Thank you for noticing this. Corrected.

Results – line 172, 176, 178

Table 3A, 3B, 3C à Table 2A, 2B, 2C

Corrected.

Discussion line 247

real-life

Corrected.

IRB statement – line 283

Feb 10, 2021

Corrected.

Table 1

All p-values need to be expressed in the correct value. For example, 0.315 or 0.021 or 0.003 or <0.001 etc.

We have changed all p-values to exact values.

And BPPS à BBPS

Corrected.

Table 2

Explanations of the abbreviation of PEG, PEG-ASC, OR, and CI need to be added in the comments below of Table 2.

Corrected.

And BPPS à BBPS

Corrected.

It would be also better if all p-values are expressed in the correct value.

We have changed all p-values to exact values.

Appendix Table

BPPS à BBPS

Corrected.

Figure 1 and Figure 2

Abbreviation of PEG and PEG-ASC need to be explained in the Figure legends.

Corrected, also for table 3 (appendix).

Round 2

Reviewer 1 Report

The authors have made a genuine effort to address my concerns. 

Minor typo here, please fix: "Nausea was more ea and vomiting were more frequent with 1L PEG-ASC compared to 2L PEG-32 ASC (p<0.001) and vomiting were to both other productssolutions (p<0.01 compared to 2L PEG-33 ASC and p<0.05 compared to 4L PEG)."

Author Response

Thank you for your quick and positive response to our changes. We have made minor additional changes in the manuscript to clarify the frequency of nausea and vomiting in the 1L PEG-ASC group compared to the other solutions. We will upload a new version of the manuscript as soon as reviewer 2 has responded.